# Antimicrobial Therapies for Early-Onset Group B Streptococcal Sepsis: Insights from an Italian Multicenter Study

**DOI:** 10.3390/antibiotics14040410

**Published:** 2025-04-16

**Authors:** Valeria Capone, Martina Buttera, Francesca Miselli, Serena Truocchio, Mattia Iaccheri, Cinzia Auriti, Roberta Creti, Lorenza Baroni, Luca Bedetti, Belinda Benenati, Giacomo Biasucci, Jenny Bua, Lidia Decembrino, Luisa Di Luca, Silvia Fanaro, Alessandra Foglianese, Lucia Gambini, Nicola Laforgia, Giuseppe Latorre, Sabrina Loprieno, Gianfranco Maffei, Lucia Marrozzini, Francesca Nanni, Giangiacomo Nicolini, Irene Papa, Barbara Perrone, Giancarlo Piccinini, Maria Rita Pulvirenti, Maria Paola Ronchetti, Enrico Rosati, Daniele Santori, Maria Eleonora Scapillati, Davide Scarponi, Sofia Spinedi, Tzialla Chryssoula, Caterina Vocale, Licia Lugli, Alberto Berardi

**Affiliations:** 1Pediatric Post-Graduate School, University of Modena e Reggio Emilia, 41125 Modena, Italy; 325566@studenti.unimore.it; 2Neonatal Intensive Care Unit, Women’s and Children’s Health Department, University Hospital of Modena, 41124 Modena, Italy; serena.truocchio@unimore.it (S.T.); bedetti.luca@aou.mo.it (L.B.); lugli.licia@aou.mo.it (L.L.); alberto.berardi@unimore.it (A.B.); 3Department of Medical and Surgical Sciences for Mother, Child and Adult, University Hospital of Modena, University of Modena and Reggio Emilia, 41124 Modena, Italy; 226966@studenti.unimore.it; 4Departmental Faculty of Medicine: General and Specialized Pediatrics, Saint Camillus International University of Health and Medical Sciences, 00131 Rome, Italy; cinzia.auriti@unicamillus.org; 5Department of Infectious Diseases, Istituto Superiore di Sanità, 00161 Rome, Italy; roberta.creti@iss.it; 6Neonatology and Neonatal Intensive Care Unit, ASMN Reggio Emilia, 42122 Reggio Emilia, Italy; lorenza.baroni@ausl.re.it; 7Pediatrics and Neonatology Unit, Guglielmo da Saliceto Hospital, 29121 Piacenza, Italy; b.benenati@ausl.pc.it (B.B.); giacomo.biasucci@unipr.it (G.B.); 8Department of Medicine and Surgery, University of Parma, 43121 Parma, Italy; 9Neonatal Intensive Care Unit, Institute for Maternal and Child Health, “IRCCS Burlo Garofolo”, 34137 Trieste, Italy; jenny.bua@burlo.trieste.it; 10ASST Pavia, Unità Operativa di Pediatria e Nido, Ospedale Civile, 27029 Vigevano, Italy; lidia_decembrino@asst-pavia.it; 11Neonatal and Pediatric Intensive Care Unit, Maurizio Bufalini Hospital, 47521 Cesena, Italy; luisa.diluca@auslromagna.it; 12Department of Medical Sciences, Pediatric Section, University of Ferrara, 44124 Ferrara, Italy; fnrslv@unife.it; 13Neonatology and Neonatal Intensive Care Unit AOUC Policlinico Bari, 70120 Bari, Italy; alessandra.foglianese@policlinico.ba.it (A.F.); nicola.laforgia@uniba.it (N.L.); annasabrina.loprieno@policlinico.ba.it (S.L.); 14Neonatal Intensive Care Unit, University Hospital of Parma, 43126 Parma, Italy; lgambini@ao.pr.it; 15Department of Interdisciplinary Medicine (DIM), University of Bari, 70120 Bari, Italy; 16Neonatology and Neonatal Intensive Care Unit, Ecclesiastical General Hospital F. Miulli, 70021 Acquaviva delle Fonti, Italy; g.latorre@miulli.it; 17Neonatology and Neonatal Intensive Care Unit, Policlinico Riuniti Foggia, 71122 Foggia, Italy; gmaffei@ospedaliriunitifoggia.it; 18Pediatric and Neonatal Unit, AUSL Modena, Ramazzini Hospital, 41012 Carpi, Italy; l.marrozzini@ausl.mo.it; 19Pediatric Unit, Ospedale Infermi, 47923 Rimini, Italy; francesca.nanni2@auslromagna.it (F.N.); irene.papa@auslromagna.it (I.P.); davide.scarponi@auslromagna.it (D.S.); 20Pediatric Unit, San Martino Hospital, 32100 Belluno, Italy; giangiacomo.nicolini@aulss1.veneto.it; 21Neonatal Intensive Care Unit, Azienda Ospedaliero Universitaria delle Marche, 60126 Ancona, Italy; barbara.perrone@ospedaliriuniti.marche.it; 22Pediatric Unit, Santa Maria Delle Croci Hospital, 48121 Ravenna, Italy; giancarlo.piccinini@auslromagna.it; 23Pediatric and Neonatal Unit, Morgagni-Pierantoni Hospital of Forlì, 47121 Forlì, Italy; ritamaria.pulvirenti@auslromagna.it; 24Neonatal Intensive Care Unit, Medical and Surgical Department of Fetus, Newborn, Infant-“Bambin Gesù” Children’s Hospital IRCCS, 00165 Rome, Italy; mariapaola.ronchetti@opbg.net; 25Neonatology and Neonatal Intensive Care Unit, 73039 Tricase, Italy; e.rosati@piafondazionepanico.it; 26Pediatric and Neonatal Unit, Azienda Ospedaliera Santa Maria degli Angeli, 33170 Pordenone, Italy; daniele.santori@asfo.sanita.fvg.it; 27Neonatal Intensive Care Unit, Hospital S.Pietro Fatebenefratelli, 00189 Rome, Italy; scapillati.eleonora@fbfrm.it; 28Neonatology and Neonatal Intensive Care Unit, Ospedale Maggiore, 40133 Bologna, Italy; sofiafiore.spinedi@ausl.bologna.it; 29Neonatal and Pediatric Unit, Polo Ospedaliero Oltrepò, ASST Pavia, 27100 Pavia, Italy; chryssoula.tzialla@unipv.it; 30Microbiology Unit IRCCS Azienda Ospedaliero, Universitaria di Bologna, 40138 Bologna, Italy; caterina.vocale@aosp.bo.it

**Keywords:** sepsis, newborn, group B streptococcus, antibiotics, antimicrobial stewardship

## Abstract

**Background**: Antimicrobial therapies used for treating group B streptococcus (GBS) early-onset sepsis (EOS) provide insight into clinicians’ adherence to antimicrobial stewardship (AMS) guidelines. **Methods**: We reviewed antimicrobial therapies given to treat newborns with GBS-EOS. Data were obtained from an Italian surveillance network (including 35 birthing centers) and were prospectively collected from 1 January 2003 to 31 December 2024. Empiric and definitive therapies were classified as adequate and inadequate. **Results**: There were 967,054 live births and 200 cases of GBS-EOS, of which 43 (21.5%) were preterm and 157 (78.5%) were full-term; 35 (17.5%) out of 200 showed no signs of illness. Fourteen (7.0%) died (one full-term and thirteen preterm newborns under 34 weeks of gestation). Based on the available information, antibiotics were adequate in 106/137 (77.4%) empiric and 48/119 (40.3%) definitive therapies. The duration of antibiotic courses did not differ between severe (median 10 days, IQR 8.0–14.0) and non-severe cases (median: 10 days; IQR: 10.0–12.5; *p* = 0.68). Antibiotic treatments lasted ≥ 15 days in 34 (20.1%) out of 169 cases with available information. **Conclusions**: In this large Italian multicenter study, deviations from international recommendations in antimicrobial therapies for GBS-EOS were critical. Our findings underscore the importance of timely antimicrobial de-escalation and the need to avoid excessively prolonged courses of antimicrobials.

## 1. Introduction

Group B streptococcus (GBS) remains a leading cause of early onset sepsis (EOS, the disease presenting from 0 to 6 days of life). GBS infections may complicate with sepsis, pneumonia, and meningitis, particularly in preterm newborns. In the USA, the current case fatality ratio is 2.1% among term infants and 19.2% among those born at <37 weeks’ gestation. The incidence of GBS EOS has decreased significantly in recent decades, thanks to effective obstetrical measures and the implementation of intrapartum antibiotic prophylaxis (IAP) [1,2].

Antibiotics are among the most frequently administered medications for newborns and are life-saving in infants with severe infection. However, the emergence of antimicrobial resistance, driven by the widespread use of antibiotics, is a global concern [3]. Antibiotics are commonly given when EOS is suspected, but the use of antibiotics in the first days of life is disproportionate compared to the current incidence of neonatal EOS. Up to 15% of all newborns receive empirical antibiotics for suspected sepsis [4], being administered annually to >200,000 US newborns at risk of EOS in the first days of life [5]. A recent large, international study compared antibiotic use during the first week of life across 13 networks in Europe, Australia, and the United States. The study compared the incidence of EOS with the burden of antibiotic treatment and reported that for each case of culture-proven sepsis, 58 newborns received antibiotics and 273 antibiotic days were administered (with the burden of treatment varying up to nine-fold across the 13 networks) [6].

Antibiotics have a potential adverse impact on the development of antimicrobial resistance (AMR) and future health [7,8]. The World Health Organization called for urgent action to avert an antimicrobial resistance crisis, focusing on antimicrobial stewardship (AMS) programs to help clinicians optimize antibiotic prescribing and improve patient outcomes. Furthermore, alterations in the development of the gut microbiome during the first few days of life condition the development of diseases in adulthood [9,10,11]. Exposure to antibiotics has been associated with an increased risk of coeliac disease, diabetes, obesity, asthma, allergies, atopic dermatitis, and others immune disorders later in life [7,8,12,13,14].

Adverse effects of perinatal antimicrobials are even more severe in the premature infant. Indeed, in this population, exposure to prolonged antibiotic treatments during the first days of life increases the risks of late-onset sepsis and necrotizing enterocolitis, both associated with unfavorable long-term neurological outcomes, and of death [15,16].

Current guidelines for EOS recommend initiating empirical antibiotic therapy for EOS with a combination of ampicillin and an aminoglycoside, such as gentamicin, to ensure effective coverage against the most common pathogens, including GBS and Escherichia coli [17,18,19,20]. This combination provides broad-spectrum coverage, with gentamicin enhancing also bacterial clearance through its synergistic effect against GBS. Following the microbiological confirmation of GBS infection, transitioning to a narrower-spectrum antibiotic (i.e., penicillin G or ampicillin) is recommended to minimize unnecessary antibiotic exposure and reduce the risk of AMR [17,18,19]. Indeed, despite the widespread use of IAP [2,21], GBS remains generally susceptible to first-line, narrow-spectrum beta-lactam antibiotics (i.e., penicillin and ampicillin) that readily cross the blood–brain barrier and are highly effective in cases of sepsis and meningitis [22,23]. Standard treatment for EOS consists of a 7–10-day course of intravenous antibiotic therapy, extended to 14–21 days in cases of uncomplicated or complicated meningitis, respectively [19,22,23,24,25]. Analyzing the antimicrobial therapies used for both the empirical and definitive treatment of GBS infections provides insight into clinicians’ adherence to antimicrobial stewardship (AMS) guidelines among healthcare centers. To assess this, we reviewed cases of GBS EOS recorded within an Italian prospective surveillance network, examining the empirical and definitive antimicrobial therapies administered in each case. Our study aims to inform centers about the antibiotic therapies administered, with the goal of encouraging adherence to international recommendations and limiting the use of unnecessary treatments.

## 2. Materials and Methods

### 2.1. Study Design

We conducted a multicenter study based on data from an Italian prospective surveillance program for GBS EOS covering 35 birthing centers [26]. Cases were included if they had a blood and/or cerebrospinal fluid culture positive for GBS within the first 72 h of life, regardless of gestational age or symptom development. A single (1 mL) or double blood culture was collected in symptomatic newborns (or in asymptomatic ones if risk factors for EOS were present; see definitions below) before initiating empirical antibiotic therapy.

Most cases (n = 164, 82%) derived from an area-based surveillance program (see Figure 1) started in 2003 in Emilia-Romagna [27], a Northern Italian region with approximately 35,000 live births (LBs) per year and 23 birthing centers. These centers are categorized as follows: level 1 (n = 12, <1000 LBs/year, inborn criteria ≥ 2000 g, ≥35 weeks); level 2 (n = 3, >1000 LBs/year, inborn criteria ≥ 1500 g, ≥32 weeks); NICUs (n = 8, no restrictions for in- and out-born newborns). All centers implement a screening-based strategy to prevent GBS EOS [28].

Local investigators extracted the following information from clinical records: demographic data, antenatal risk factors, clinical status at birth, investigations, and antibiotics received. Specifically, maternal data included prenatal screening for GBS, mode of delivery, presence of GBS risk factors (maternal fever during labor, GBS bacteriuria during pregnancy, previous infant with invasive GBS infection, rupture of membranes ≥ 18 h, preterm birth), and any intrapartum antibiotic prophylaxis. Neonatal data included sex, birth weight, clinical symptoms (if present), laboratory data, pathogen identification, antibiogram, empiric/definitive antibiotic therapy administered, results of instrumental examinations (MRI or cerebral ultrasound), infection severity, and outcome.

To ensure patient confidentiality, spreadsheets submitted to the principal investigator were fully anonymized and did not include any identifiable information about patients or caregivers. Any missing data were retrieved through direct phone contact with center representatives to maximize data completeness. To minimize information loss, a monthly email was sent to designated pediatricians/neonatologists and laboratory representatives of participating centers to request case notifications. Using the available data, we assessed the duration and appropriateness of antibiotic therapy (see below) and correlated the administered treatments with the severe outcomes of infection.

The study’s starting year coincides with the establishment of the GBS surveillance network in Emilia-Romagna. According to data provided from the Regional Health Agency, the number of LBs in the Emilia-Romagna region from 2003 to 2024 was 784,785 (the estimate for 2024 is not yet available, and the total number was calculated assuming a number of live births for that year comparable to 2023).

Starting in 2016 (see Figure 1), centers outside the Emilia-Romagna region (12 centers located in Northern, Central, and Southern Italy; level 1: n = 2; level 2: n = 1; NICUs: n = 9) increasingly began reporting cases of GBS EOS to the network. Based on the estimates provided by the referring clinician of each individual center, the overall number of LBs in these 12 birthing centers outside Emilia-Romagna was 182,269. Since the out-of-region centers initially recorded the cases inconsistently, we cannot provide an accurate epidemiological estimate. Thus, an accurate and area-based count of LBs is only available for the Emilia-Romagna region, and we provide GBS EOS incidence data exclusively for this region. This study analyzes antimicrobial treatments of all cases of GBS EOS that were entered into the network database, coming from birth centers within or outside the Emilia-Romagna region.

We reviewed in detail all cases prospectively registered in the network from 1 January 2003 to 31 December 2024. For each proven episode, the antibiotic treatment (compliance with antibiotic recommendations for length of antibiotic therapy, duration of each antimicrobial treatment, choice of antimicrobials and reassessment of antibiotic therapy at 72 h) was reviewed in detail by two investigators (MB, VC). In case of disagreement, a third investigator (AB) reviewed the cases and assign them to either an appropriate or inappropriate treatment.

The project was approved by the local ethics committee (protocol no. 910/2020/OSS/AOUMO, amended in 2024). Due to the inability to retrospectively obtain consent for participation in the study during the early years of surveillance, the ethics committee waived the requirement for informed consent. This study followed the Strengthening the Reporting of Observational Studies in Epidemiology (STROBE) guidelines.

### 2.2. Definitions

**EOS case**: isolation of GBS from blood or cerebrospinal fluid (CSF) culture or through polymerase chain reaction (PCR) on CSF obtained within the first 72 h of life [29,30,31].

**Risk factors** (RFs) include preterm birth (<37 weeks’ gestation), a previous infant with GBS invasive infection, rupture of membranes 18 h prior to delivery (ROM), GBS bacteriuria identified during the current pregnancy, and maternal intrapartum fever 38 °C during labor.

**Adequate intrapartum antibiotic prophylaxis (IAP)**: Administration of penicillin, ampicillin, or cefazolin at least 4 h before delivery [28].

**Adequacy of empiric neonatal therapy**: Administration of penicillin or ampicillin, with an aminoglycoside (± a third-generation cephalosporin), within the first 48–72 h following blood culture collection [17,18,19,20,22].

**Adequacy of definitive neonatal therapy (or de-escalation)**: Administration of penicillin or ampicillin after 48–72 h of empiric therapy [17,18,19,20,22].

**Asymptomatic (occult) bacteremia**: A positive blood culture obtained due to maternal risk factors for EOS in an infant who remains asymptomatic.

**Culture-proven meningitis**: A GBS-positive cerebrospinal fluid (CSF) culture (or PCR test) in infants presenting with clinical signs of meningitis.

**Brain lesions**: Any brain lesion identified and confirmed via ultrasound or MRI.

**Severe disease**: Defined by the presence of any of the following: need for catecholamine support, mechanical ventilation, seizures, meningitis, brain lesions at hospital discharge, or death [26].

### 2.3. Statistical Analysis

Statistical analysis was performed using MedCalc version 9.3 (MedCalc Software, Ostend, Belgium). Continuous variables are expressed using mean and standard deviation (SD) or median and interquartile range (IQR); binary and categorical data are reported as frequencies and percentages. The Mann–Whitney rank sum test was used for unadjusted comparisons of continuous variables between groups; Pearson’s χ^2^ test and Fisher’s exact test (when one or more cells had expected frequency < 5) were used for unadjusted comparisons of categorical variables between groups. A *p*-value < 0.05 was considered statistically significant.

## 3. Results

### 3.1. Study Population

During the study period, the network registered 967,054 LBs and 200 cases of GBS EOS in 35 birthing centers. Of these 200 cases, 164 (82.0%) occurred in Emilia-Romagna (live births: 784,785; incidence rate of 0.21 per 1000). The majority of affected newborns were full-term and male (Table 1).

Compared to full-term infants, preterm newborns were more likely to be delivered via cesarean section, be born to mothers without intrapartum fever, have a positive vaginal-rectal swab for GBS, and experience prolonged rupture of membranes. Less than 25% of the newborns had been exposed to intrapartum antibiotic prophylaxis (IAP), which was adequate in less than 20% of cases.

### 3.2. Clinical Findings

Table 2 displays clinical characteristics of the study population. Nearly 20% of the newborns were asymptomatic, fewer than 7% developed brain lesions, but 47% experienced a severe disease, 6% had meningitis, and 7% of newborns died from GBS EOS (the age at death is reported in the footnote of the table). As compared to preterm, full-term newborns were less likely to undergo mechanical ventilation, to receive catecholamine support, to experience a severe disease, and to die after GBS EOS. Among the fourteen newborns who died, only one (0.5% of the overall population) was born at term, whereas the remaining 13 were born preterm with a gestational age under 34 weeks.

### 3.3. Microbiological Results

Among the 200 newborns with GBS EOS, 186 (93.0%) had only a GBS positive blood culture, 6 (3%) had only GBS positive CSF culture (or CSF PCR), while the remaining 8 (4%) had both positive blood and CSF culture (and/or CSF PCR). However, only 103 newborns (51.5%) underwent a lumbar puncture.

### 3.4. Overall Antibiotic Exposure, Empirical and Definitive Antimicrobial Therapies

Among the 200 newborns with GBS EOS, 6 (3%) did not receive any antibiotic treatment (because they remained asymptomatic or had suffered mild and transient symptoms in the very first hours of life, becoming asymptomatic when the blood culture results were available), whereas 194 (97%) were treated with antibiotics. Figure 2 displays the most commonly used antibiotic combinations for both empiric and definitive therapy. Overall, more than half of the newborns received a combination of ampicillin and gentamicin. However, a significant proportion of newborns received only ampicillin (or penicillin) throughout the entire treatment course, without ever adding aminoglycosides, as is typically recommended for empiric therapy.

Among the 194 newborns treated with antibiotics, 8 died within the first 72 h of life and were excluded from the analysis of appropriate empirical therapies. Of the remaining 186 newborns, data were unavailable for 49 cases, leaving 137 newborns for analysis. Appropriate empirical antibiotic coverage was administered to 106 (77.4%) of these 137 newborns. The remaining 31 newborns (22.6%) received antibiotics not typically recommended for empirical therapy in EOS, such as ampicillin–sulbactam, piperacillin, teicoplanin, vancomycin, and clarithromycin or only penicillin/ampicillin.

Definitive therapy was administered with appropriate antibiotic coverage in only a minority of cases (40%), although information was available only for 119 (59.5%) of the 200 newborns.

Figure 3 illustrates the reasons for the inappropriateness of antimicrobial therapy. The primary issue identified was the extension of initial empirical therapy with aminoglycosides beyond the recommended 72 h period; notably, in nine newborns, aminoglycosides were discontinued within 96 h. Another issue was the use of excessively broad-spectrum antimicrobials, such as amoxicillin–clavulanic acid, ampicillin–sulbactam, meropenem, ceftazidime, and cefotaxime.

The duration of antibiotic therapy was calculated excluding the deceased newborns (n = 14), who received antibiotics for varying durations (0 to 12 days) until death, and the untreated cases (n = 6). Data on antibiotic therapy duration were available for 168 (84%) of the remaining 180 survivors who received antibiotics. Among these, 30 infants received treatment lasting seven days or less (with 14 receiving two to six days). None of these infants experienced a recurrence of GBS infection. By contrast, 50 (25%) of 168 were treated for more than 14 days. The overall median duration of antibiotic therapy was 10 days (IQR 9.0–14.0). Duration did not differ significantly between full-term (median 10 days, IQR 8–14) and preterm newborns (median 10 days, IQR 8.7–18.0; *p* = 0.28). The duration of antibiotic therapy was longer in newborns who had suffered from meningitis (median 16 days, IQR 14.0–20.3) compared to those without meningitis (median 10 days, IQR 10.0–14.0, *p* < 0.01).

### 3.5. Antibiotic Exposure in Asymptomatic and Severely Ill Newborns

Information on antibiotic therapy was missing for 14 out of 35 asymptomatic newborns. Among the remaining 21, 3 (8.5%) did not receive any treatment, while the other 18 underwent antibiotic courses of varying durations. Excluding missing cases and those who died, the duration of treatment was significantly longer in symptomatic newborns (median 10 days, IQR 10–14.5) compared to asymptomatic newborns (median 10 days, IQR 6.3–10.0; *p* < 0.001).

Among the 56 newborns with severe disease, 14 died. In the remaining 42 surviving newborns, the median duration of antibiotic therapy was 10 days (IQR 8.0–14.0). No difference was observed in treatment duration between newborns with severe disease (median 10 days, IQR 8.0–14.0) and those with non-severe disease (median 10 days, IQR 10.0–12.5; *p* = 0.68).

## 4. Discussion

The current study provides an overview of antimicrobial (including empirical and definitive) therapies administered in Italian network centers to treat GBS EOS during the first days of life. To our knowledge, this is the first study in a large catchment area to analyze the appropriateness of antibiotic prescriptions for GBS EOS. The analysis of these data allows us to assess clinicians’ adherence to recommendations for treating newborns, with the aim of promoting compliance with international guidelines and reducing the use of unnecessary treatments.

Within our network, less than 25% of the newborns with GBS EOS had been exposed to IAP, which was adequate in only 5% of cases. These findings underscore the importance of adhering to established guidelines for GBS screening and IAP administration to effectively reduce the incidence of GBS EOS [2,21].

Although some infants had a non-severe infection (with ~18% of the infants remaining asymptomatic), nearly half (46.7%) of the newborns developed severe disease (particularly those preterm born). Although fewer than 7% of infants developed brain lesions at discharge from hospital, their long-term neurodevelopmental outcomes remain a concern [32]. Case fatalities affected 7% of cases. We cannot determine how much of the mortality was related to extreme prematurity itself (especially in those who died more than seven days after onset) as opposed to GBS infection. Overall, these findings reinforce the critical need for continued improvements in perinatal infection prevention, early recognition, and tailored management strategies, particularly for preterm newborns who remain the most vulnerable group.

Despite the low rates of lumbar punctures performed (and the potential underestimation), a few meningitis cases (6%) were confirmed through CSF culture or PCR. While lumbar puncture remains a crucial diagnostic tool [8,33], bacterial meningitis occurred in a smaller subset of newborns with GBS EOS. This could reflect an underestimation of the number of meningitis cases due to the initiation of antibiotic therapy before performing the lumbar puncture; alternatively, it could reflect a lower incidence of GBS meningitis (compared to the early studies on GBS infections) because of widespread prevention. Indeed, a significant reduction in the incidence of both GBS EOS and meningitis has been reported over the past two decades after GBS prevention [27,34].

Due to their asymptomatic presentation, 3% of newborns remained untreated. The close observation of these newborns is essential to ensure no late-onset complications arise [35]. Regional surveillance in Emilia Romagna also includes late-onset GBS infections, and there is no evidence that the lack of antibiotic treatment for asymptomatic cases has had an impact on the occurrence of late-onset infections. By contrast, the majority (97%) of newborns received antibiotic treatment, with more than half receiving the guideline-recommended empirical combination of ampicillin and gentamicin.

Although targeted interventions in antibiotic duration and selection could further optimize neonatal sepsis management, a significant number of empirical therapies deviated from recommended guidelines [17,18,19,20,21,22]. Some newborns were treated with ampicillin (or penicillin) alone, without the addition of gentamicin. While GBS remains highly susceptible to beta-lactams, omitting gentamicin in the empiric phase may reduce the speed of bacterial clearance, particularly in severe cases [36]. The preference for monotherapy may reflect institutional protocols, concerns about aminoglycoside toxicity (e.g., nephrotoxicity or ototoxicity), or an assessment of lower disease severity in certain newborns. Sometimes, monotherapy may be justified by starting antibiotic therapy only after knowing the results of the blood culture, as is typically the case with asymptomatic bacteremia. However, current guidelines generally recommend a combination therapy until microbiological confirmation allows for de-escalation [22,24].

Definitive therapies were appropriate in only a minority of cases, although this information may be partly biased by a high number of missing data. Aminoglycosides were sometimes administered beyond the recommended 72 h period. Their extended use without a clear indication may increase the risk of nephrotoxicity and ototoxicity. Of note, in seven newborns, aminoglycosides were discontinued within 96 h, thus slightly exceeding the recommended duration. Although this extension may have had minimal clinical impact, it suggests a potential delay in definitive therapy adjustments based on microbiological results. Furthermore, the administration of amoxicillin–clavulanic acid, ampicillin–sulbactam, meropenem, ceftazidime, and cefotaxime in some cases indicates a tendency toward overly broad antimicrobial coverage. While these antibiotics may be necessary in specific cases (e.g., polymicrobial infections, antibiotic resistance concerns, or severe sepsis with unidentified pathogens), their routine use in GBS infections is not recommended, and beta-lactams (i.e., penicillin or ampicillin) remain the first-line treatment for confirmed GBS infections [19,22].

The duration of antibiotic therapy in newborns with GBS infections should be guided by the clinical presentation, response to treatment, and microbiological findings, aiming to avoid prolonged antibiotic use unless clinically justified. This approach ensures the complete eradication of the infection and prevents potential complications [19]. Guidelines suggest a treatment course of 7-to-10 days for sepsis, whereas in cases of GBS meningitis, the duration of antibiotic treatment should be no less than 14 days, with appropriate dosing to ensure effective penetration into the cerebrospinal fluid [17,18,19,22,24]. However, evidence-based data to definitively recommend the optimal duration of antimicrobial therapy for neonatal sepsis or meningitis (due to GBS or other pathogens) remain insufficient [37,38,39]. A recent nationwide study in Sweden encompassing 1,025,515 newborns and 647 cases of EOS reported a median treatment duration of 8 days for EOS [40]. In cases of neonatal bacterial meningitis, studies have explored shorter treatment durations. One randomized controlled trial found that a 10-day course of parenteral antibiotics was as effective as a 14-day regimen, with no significant difference in mortality rates [41].

In our study, the overall median duration of antibiotic therapy was 10 days, with no significant differences between preterm and full-term newborns. However, symptomatic newborns received longer treatment than asymptomatic ones, regardless of symptom severity. The observation that 20% of newborns received antibiotic therapy lasting 15 days or more, significantly higher than the diagnosed cases of meningitis in this population—raises questions about the appropriateness of the treatment duration in certain patients. Extending therapy beyond recommended periods should be justified by specific clinical indications or documented complications.

Our study has important limitations. The lack of detailed information on the clinical criteria or biomarkers used to prolong antimicrobial therapy in these cases limits our ability to assess adherence to guidelines and the appropriateness of therapeutic decisions. Possible factors contributing to inappropriate antibiotic use include variability in clinical practices, a lack of awareness of guidelines, or resource limitations. We were unable to retrieve the burden of these contributing factors. Furthermore, we lack detailed information regarding clinical circumstances of death.

Overall, our findings suggest that, despite recommendations for tailoring the antibiotic therapy duration based on clinical severity and individual neonatal response, actual clinical practice tends to follow a uniform approach, regardless of disease presentation. This may reflect a cautious strategy by clinicians to ensure adequate antimicrobial coverage against potential pathogens. However, it also underscores the need for more precise guidelines and ongoing medical education to optimize antibiotic use in neonatal care and minimize the risk of developing antimicrobial resistance and side effects associated with extended antibiotic treatments.

## 5. Conclusions

In conclusion, antibiotics have a potential adverse impact on the development of antimicrobial resistance (AMR) and future health. Our findings underscore the importance of timely antimicrobial de-escalation and the need to minimize unnecessary broad-spectrum antibiotic use. While most newborns received appropriate therapy, targeted interventions in antibiotic duration and selection could further optimize neonatal sepsis management, reduce AMR risks, and improve overall neonatal outcomes. Our study informs centers about the antibiotic therapies administered, with the goal of encouraging adherence to international recommendations and limiting the use of unnecessary treatments. In the future, an additional goal could be the creation of an antibiotic stewardship team that fosters shared decision making regarding intrapartum antibiotics and the initiation or discontinuation of neonatal antibiotic therapies, as recommended in the literature.

## Figures and Tables

**Figure 1 antibiotics-14-00410-f001:**
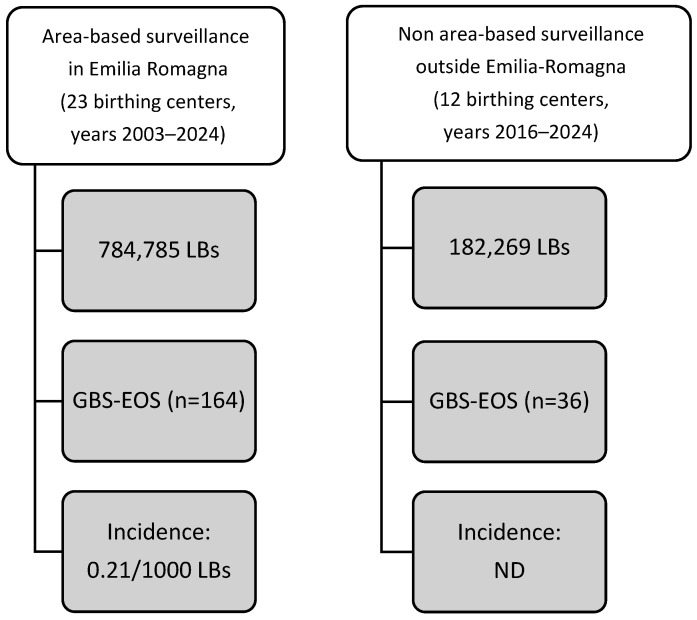
Sources of GBS cases and surveillance network. EOS, early-onset sepsis; GBS, group B streptococcus; LBs, live births; ND, not determined.

**Figure 2 antibiotics-14-00410-f002:**
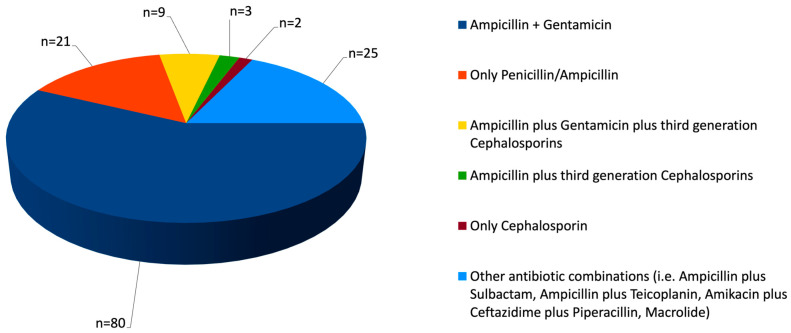
Empiric and definitive antibiotic regimens; cases with missing information (n = 40), case fatalities (n = 14), and newborns who remained untreated (n = 6) were excluded.

**Figure 3 antibiotics-14-00410-f003:**
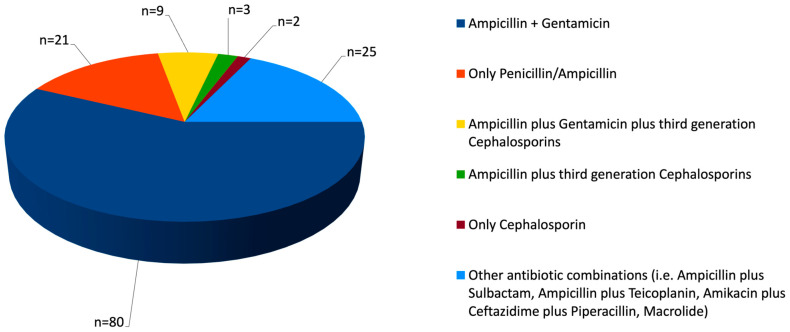
Main reasons for inadequate definitive therapies. One case initially received ampicillin that was then followed by the addition of meropenem; the case was considered adequate since the newborn had severe complications (leading to intestinal perforation).

**Table 1 antibiotics-14-00410-t001:** Demographics and clinical data of newborns with early-onset group B streptococcus.

	All (n = 200)	Preterm Newborns (n = 43)	Missing	Full-Tem Newborns (n = 157)	Missing	*p*
Gestational age, wks, median (IQR)	39 (37–40)	30 (26–35)	0	40 (38–40)	0	NA
Birth weight, gr, median, (IQR)	3200 (2777.50–360.00)	1530 (863.75–2539.75)	0	3352 (3053.75–3677.50)	0	NA
Male sex, n (%)	108 (54.0)	28 (65.1)	0	80 (50.9)	0	0.14
Vaginal delivery, n, (%)	141 (70.8)	21 (48.8)	0	120 (76.9)	1	<0.01
Vaginal-rectal screening swab performed, n, (%)	158 (79.4)	19 (45.2)	1	139 (88.5)	0	0.96
Positive vaginal-rectal screening swab, n (%)	64 (40.2)	14 (73.7)	24	50 (35.7)	17	<0.01
IAP, n, (%) †	47 (24.0)	14 (34.1)	2	33 (21.3)	2	0.13
Obstetrical RFs for EOS, n, (%)						
Previous infant with GBS sepsis	1 (0.5)	1 (2.4)	1	0 (0%)	3	0.48
GBS bacteriuria	21 (11.8)	7 (18.9)	6	14 (9.9)	16	0.22
Maternal fever during labor (>38 °C)	35 (17.5)	2 (4.7)	0	33 (21.0)	0	0.03
ROM > 18 h	26 (13.4)	11 (26.8)	2	15 (9.8)	4	<0.01

EOS, early-onset sepsis; GBS, group B streptococcus; IAP, intrapartum antibiotic prophylaxis; IQR, interquartile range; NA, not applicable; RFs, risk factors; ROM, rupture of membranes. † adequate IAP: ampicillin, penicillin, cefazolin given >4 h before delivery. IAP was adequate in 9 (19.1%) of 47 cases exposed to IAP (preterm newborns, n = 4; term newborns, n = 5).

**Table 2 antibiotics-14-00410-t002:** Clinical findings of newborns with early-onset group B streptococcus disease.

	All (n = 200)	Preterm Newborns (n = 43)	Missing	Full-Term Newborns (n = 157)	Missing	*p*
Mechanical ventilation, n, (%)	38 (19.6)	20 (48.8)	2	18 (11.8)	4	<0.01
Catecholamines, n, (%)	31(16.0)	14 (34.1)	2	17 (11.1)	4	<0.01
Meningitis, n (%) †	12 (6.0)	0 (0)	0	12 (7.6)	0	0.31
Brain lesions, n (%)	12 (6.5)	3 (10.0)	0	9 (5.8)	1	0.65
Seizures, n (%)	10 (6.5)	3 (8.6)	8	7 (5.8)	37	0.85
Asymptomatic, n (%)	35 (17.5)	3 (7)	0	32 (20.4)	0	0.07
Case fatalities, n (%) §	14 (7.0)	13 (30.2)	0	1 (0.6)	1	<0.01

† Based on positive CSF culture (n = 8), polymerase chain reaction (n = 3) or both (n = 1). ¶ Newborns who died were excluded from the calculation. § among 14 case fatalities, 5 died within 24 h from the onset of EOS, 6 from 1 to 7 days and 3 over the first week.

## Data Availability

The raw data supporting the conclusion of this article will be made available by the authors on request.

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
