# Peer review of "Antimicrobial Therapies for Early-Onset Group B Streptococcal Sepsis: Insights from an Italian Multicenter Study"

_antibiotics, 2025, doi:10.3390/antibiotics14040410_

Round 1

Reviewer 1 Report

Comments and Suggestions for Authors

Introduction

Line 77-79 please explain the fatality rate of GBS.

Please need to explain the sources and subtype of GBS. 

Many information is missing in introduction. Please explain clearly. 

Table 1 number of N should be increase. It is very less to say a novel findings. Thats why many data showed no statistical significant. 

Positive vaginal-rectal screening swab why too many number is missing ?

It would be more interesting if you add fiscal year. 

Also you can add some seasonal birth for ex Summer or winter. 

To be honest, there is no interesting or novel findings in your research. 

You should add maternal and infant characteristics. 

Discussing need more modification. It looks like you explain again results section. 

Line 351-364 and 379-389 no need too much explanation your result. Actually in the discussion part you should add your result compare with other findings. 

Total 200 cases is very small amount please increase the cases number. 

Comments on the Quality of English Language

ok

Author Response

Response to Reviewer 1 Comments

Comments 1: Line 77-79 please explain the fatality rate of GBS.

Response 1: Changed as suggested

Comments 2: Please need to explain the sources and subtype of GBS

Response 2: Our study investigates antibiotic treatments for early-onset GBS infections, which are invariably of maternal origin (as is well known by those working in the field of neonatal GBS infections). Information on the serotypes has already been published elsewhere (see DOI: 10.3390/microorganisms9122579).

Comments 3: Many information is missing in introduction. Please explain clearly.

Response 3: Please clarify more clearly what the reviewer is referring to.

Comments 4: Table 1 number of N should be increase. It is very less to say a novel findings. Thats why many data showed no statistical significant.

Response 4: We do not fully understand this comment. For authors familiar with this field, our population is unusually large; furthermore, we are unaware of previous studies that have examined in detail antibiotic therapies in early-onset GBS infection. If the reviewer is aware of any, please let us know.

Comments 5: Positive vaginal-rectal screening swab why too many number is missing ?

Response 5: Antenatal screening (whose results are available at the time of delivery) are performed at ≥ 35 weeks gestation; in contrast, screening is not performed under 35 weeks’ gestation, and information on GBS carriage may be unavailable at the time of delivery. In this case as well, for those familiar with neonatal GBS infections, this data is not surprising.

Comments 6: It would be more interesting if you add fiscal year.

Response 6: Please, explain what is “fiscal year” and what should we refer to?

Comments 7: Also you can add some seasonal birth for ex Summer or winter.

Response 7: Sorry, Group B Streptococcus is not a seasonal infection

Comments 8: To be honest, there is no interesting or novel findings in your research. 

Response 8: We have nothing further to add.

Comments 9: You should add maternal and infant characteristics.

Response 9: See table 1

Comments 10: Discussing need more modification. It looks like you explain again results section.

Response 10: The discussion has been revised, also in accordance with the suggestions of the other reviewers

Comments 11: Line 351-364 and 379-389 no need too much explanation your result. Actually Response 11: in the discussion part you should add your result compare with other findings. The discussion has been revised, also in accordance with the suggestions of the other reviewers.

Comments 12: Total 200 cases is very small amount please increase the cases number.

Response 12: This comment suggests a lack of familiarity with neonatal GBS infections

Reviewer 2 Report

Comments and Suggestions for Authors

The manuscript provides limited information on the statistical methods used. Include a more comprehensive description of the statistical analyses performed. Specify any statistical tests used, definitions of significance levels, and whether adjustments were made for confounding variables. While the use of non-parametric tests (Mann-Whitney) and Chi-square tests is appropriate, more details about how continuous variables. If any adjustments for multiple comparisons were performed, please state this explicitly. The manuscript mentions that data were missing for a number of cases. Please provide a more detailed description of the extent of missing data and any sensitivity analyses or imputation methods used.

The criteria for classifying therapies as 'adequate' or 'inadequate' are briefly mentioned but lack detailed justification. Elaborate on the specific guidelines or consensus statements used to define adequacy of therapy. Including references to international standards will strengthen the rationale behind the categorizations.

While the study identifies deviations from AMS guidelines, it does not thoroughly explore the underlying reasons. Expand the discussion to consider possible factors contributing to inappropriate antibiotic use, such as variability in clinical practices, lack of awareness of guidelines, or resource limitations. The limitations of the study are not fully addressed.

The manuscript highlights problems but offers limited actionable solutions. Offer specific recommendations for improving adherence to AMS guidelines. 

Author Response

Response to Reviewer 2 Comments

Comments 1: The manuscript provides limited information on the statistical methods used. Include a more comprehensive description of the statistical analyses performed. Specify any statistical tests used, definitions of significance levels, and whether adjustments were made for confounding variables. While the use of non-parametric tests (Mann-Whitney) and Chi-square tests is appropriate, more details about how continuous variables. If any adjustments for multiple comparisons were performed, please state this explicitly. The manuscript mentions that data were missing for several cases. Please provide a more detailed description of the extent of missing data and any sensitivity analyses or imputation methods used.

Response 1: We thank the Reviewer for this insightful comment. In response, we have expanded the “Statistical Analysis” section to provide a more comprehensive description of the methods used. Specifically, we have now detailed the statistical tests applied, including the use of non-parametric tests (Mann–Whitney U test) for continuous variables and Chi-square or Fisher’s exact tests for categorical variables, as appropriate. We have also clarified how continuous and categorical variables are represented in the manuscript (i.e., medians and interquartile ranges for continuous variables; counts and percentages for categorical variables). Furthermore, we have explicitly stated that all comparisons were unadjusted. No adjustments for multiple comparisons were performed, nor for missing data. However, a detailed description of the burden of missing data for each variable is reported in the text

Comments 2: The criteria for classifying therapies as 'adequate' or 'inadequate' are briefly mentioned but lack detailed justification. Elaborate on the specific guidelines or consensus statements used to define adequacy of therapy. Including references to international standards will strengthen the rationale behind the categorizations.

Response 2: We added a new reference (Fuchs, WHO 2018), including sources on the adequacy of antibiotic therapies, also in the definitions section (see Methods).         

Comments 3: While the study identifies deviations from AMS guidelines, it does not thoroughly explore the underlying reasons.

Expand the discussion to consider possible factors contributing to inappropriate antibiotic use, such as variability in clinical practices, lack of awareness of guidelines, or resource limitations. The limitations of the study are not fully addressed.

Response 3: For both comments, we added these limitations, as suggested

Comments 4: The manuscript highlights problems but offers limited actionable solutions. Offer specific recommendations for improving adherence to AMS guidelines.

Response 4: We thank the reviewer for this comment. However, a practical aspect of our study is to inform all centers about the antibiotic therapies administered and the opportunity to follow international recommendations to limit the use of unnecessary therapies. This aim has been added in Introduction and Conclusions. In the future, an additional goal could be the creation of an antibiotic stewardship team that fosters shared decision-making regarding intrapartum antibiotics and the initiation or discontinuation of neonatal antibiotic therapies, as recommended in the literature.

Reviewer 3 Report

Comments and Suggestions for Authors

Empirical and Definitive Antimicrobial therapies in Early-Onset Group B Streptococcal Infection: Findings from an Italian Network

Valeria Capone 1,*, Martina Buttera 1,*, Francesca Miselli 2,*, Serena Truocchio 2 , Mattia Iaccheri 3 , Cinzia Auriti 4 , 5 Roberta Creti 5 , Lorenza Baroni 6 , Luca Bedetti 2 , Belinda Benenati 7 , Giacomo Biasucci 7,8, Jenny Bua 9 , Lidia 6 Decembrino 10, Luisa Di Luca 11, Silvia Fanaro 12, Alessandra Foglianese 13 , Lucia Gambini 14 , Nicola Laforgia 13,15 , 7 Giuseppe Latorre 16, Sabrina Loprieno 13, Gianfranco Maffei 17, Lucia Marrozzini 18, Francesca Nanni 19, Giangiacomo 8 Nicolini 20, Irene Papa 19, Barbara Perrone 21, Giancarlo Piccinini 22, Maria Rita Pulvirenti 23, Maria Paola Ronchetti 24 , 9 Enrico Rosati 25 , Daniele Santori 26 , Maria Eleonora Scapillati 27 , Davide Scarponi 19, Sofia Spinedi 28 , Tzialla 10 Chryssoula 29 , Caterina Vocale 30, Licia Lugli 2 and Alberto Berardi 2,*

Minor revision

I appreciate all  the authors for the efforts they put in this clinical epidemiological  study on GBS EOS with references to antimicrobial resistance and the importance of antimicrobial stewardship. This study evaluates the adequacy of empirical and definitive antimicrobial therapies for early-onset Group B Streptococcus (GBS) sepsis in newborns across 35 Italian birthing centers. It aims to assess adherence to antimicrobial stewardship (AMS) guidelines, analyze treatment duration, and examine the relationship between therapy adequacy and clinical outcomes. The study covers 200 cases of GBS EOS over a substantial period (2003–2024) providing a large sample size for robust analysis. The research highlights critical deviations from international antimicrobial stewardship guidelines, emphasizing the need for better adherence. The study adheres to the STROBE guidelines and uses clear definitions for key terms (e.g., adequate therapy, severe disease). Clear presentation of statistical methods (Mann-Whitney and Chi-square tests) enhances the validity of the findings.

I would like to give my insights that would better explain the objective of the study to the wider audience.

  1. I would like to suggest, considering a more concise and engaging title that audience would engage easily.

Title of suggestion: “Antimicrobial Therapies for Early-Onset Group B Streptococcal Infection: Insights from an Italian Multicenter Study."

  1. Abstract:
  2. Typos highlighted in red
  • (GBS) early-onset sepsis (EOS) provides insight into clinicians' adherence to antimicrobial
  • whereas 165 (82.5%) developed symptoms (56 of them - 0%- had severe disease) 62
  • were preterm newborns under 34 weeks gestation). Based on the available information, 64
  1. The results section of the abstract is very dense. break it down for easier reading. You could separate the incidence, adequacy of therapy and mortality

iii. Lines70,71 - Our findings underscore the importance of timely antimicrobial de-escalation and the need to avoid excessively prolonged courses of antimicrobials. - mention the implications of inappropriate antibiotic use in the conclusion.

  1. The study period is from 2003–2024,explain why this study period is chosen and how it reflects changing clinical practices on GBD EOS in the introduction.
  2. Discuss bias of selection criteria of Emilia-Romagna and the underrepresentation of centers outside Emilia-Romagna
  3. Clarify how missing data may bias the findings, especially regarding adequacy and duration of antibiotic treatment.
  4. Highlight statistically significant results using bold text or asterisks in Table 1 & Table 2

7.Why some cases received extended antibiotic courses and what biomarkers considered?

8.Why there were 14 deaths could you provide the clinical context for it?

9.In the discussion expand your comparison with recent European and North American research on neonatal sepsis management. Also avoid long and complex sentences.

Limitations of study:

Biomarker data and its impact on assessing therapy is adequate

Observed inter-center variability in antibiotic protocols.

Recommendations: Manuscript can be considered for publication after minor revisions

Author Response

Response to Reviewer 3 Comments

3. Point-by-point response to Comments and Suggestions for Authors

Comments 1:

·      Abstract:

Typos highlighted in red

•          (GBS) early-onset sepsis (EOS) provides insight into clinicians' adherence to antimicrobial

•          whereas 165 (82.5%) developed symptoms (56 of them - 0%- had severe disease) 62

•          were preterm newborns under 34 weeks’ gestation). Based on the available information, 64

The results section of the abstract is very dense. break it down for easier reading. You could separate the incidence, adequacy of therapy and mortality.

Response 1:

To improve reading, we summarized some results, as suggested. However we cannot add the overall incidence for the reasons reported in Methods

Comments 2:

Mention the implications of inappropriate antibiotic use in the conclusion.

Response 2:

Mentioned, as suggested.

Comments 3:

The study period is from 2003–2024, explain why this study period is chosen and how it reflects changing clinical practices on GBD EOS in the introduction” and “Discuss bias of selection criteria of Emilia-Romagna and the underrepresentation of centers outside Emilia-Romagna”

Response 3:

We clarified in Methods that the study's starting year coincides with the establishment of the GBS surveillance network in Emilia-Romagna. Since 2016, we have expanded the surveillance to include centers outside the Emilia-Romagna region. Given that 82% of the cases come from the Emilia-Romagna region, we do not believe that cases from outside the region will substantially distort the results.

Comments 4:

Clarify how missing data may bias the findings, especially regarding adequacy and duration of antibiotic treatment.

Response 4:

In response, we have expanded the “Statistical Analysis” section to provide a more comprehensive description of the methods used. Specifically, we have now detailed the statistical tests applied, including the use of non-parametric tests (Mann–Whitney U test) for continuous variables and Chi-square or Fisher’s exact tests for categorical variables, as appropriate. We have also clarified how continuous and categorical variables are represented in the manuscript (i.e., medians and interquartile ranges for continuous variables; counts and percentages for categorical variables). Furthermore, we have explicitly stated that all comparisons were unadjusted. No adjustments for multiple comparisons were performed, nor for missing data. However, a detailed description of the burden of missing data for each variable is reported in the text.

Comments 5:

Highlight statistically significant results using bold text or asterisks in Table 1 & Table 2

Response 5:

Thank you, but this suggestion does not match with journals’ guidelines

Comments 6:

Why some cases received extended antibiotic courses and what biomarkers considered?

Response 6:

This is a major limitation of our study We added in the Discussion: Our study has important limitations. The lack of detailed information on the clinical criteria or biomarkers used to prolong antimicrobial therapy in these cases limits our ability to assess adherence to guidelines and the appropriateness of therapeutic decisions. Possible factors contributing to inappropriate antibiotic use, such as variability in clinical practices, lack of awareness of guidelines, or resource limitations. We were unable to retrieve the burden of these contributing factors.

 Comments 7:

Why there were 14 deaths could you provide the clinical context for it?

 Response 7:

Available information regarding causes of death is insufficient, and we added this limitation in the discussion. Furthermore, we added in the footnote of table 2: “among 14 case fatalities, 5 died within 24 hours from the onset of EOS, 6 from 1 to 7 days and 3 over the first week”

Comments 8:

In the discussion expand your comparison with recent European and North American research on neonatal sepsis management. Also avoid long and complex sentences.

Response 8:

Updated both in the Introduction and in Methods the reference to guidelines

Comments 9: Limitations of the study “Biomarker data and its impact on assessing therapy” and “Observed inter-center variability in antibiotic protocols”

Response 9: Updated (see limitations)

Reviewer 4 Report

Comments and Suggestions for Authors

The manuscript is well written. However, I have a few comments to improve the manuscript:

  1. In the manuscript is everywhere EOS (sepsis), while in the title is infection. I suggest to write in the title as well sepsis instead of infection, since is not the same.
  2. Line 131: "risk factors for EOS present" could you give examples here?
  3. Line 229: Please give the explanation for LP abbreviation.
  4. Lines 404-407: " While the majority of new-borns received appropriate therapy, targeted interventions in antibiotic duration and selection could further optimize neonatal sepsis management, reduce AMR risks, and improve overall neonatal outcomes. " I would interested in whether the appropriate (guideline adherent) therapy led to better clinical outcomes or not. Have you made some analysis on this? Introducing result regarding the effect of ASP on clinical outcomes in EOS would increase the quality of the manuscript.

Author Response

Response to Reviewer 4 Comments

Point-by-point response to Comments and Suggestions for Authors

Comments 1: In the manuscript is everywhere EOS (sepsis), while in the title is infection. I suggest to write in the title as well sepsis instead of infection, since is not the same.

Response 1: Changed, as suggested

Comments 2:  "risk factors for EOS present" could you give examples here?

Response 2:  Added in Methods, as suggested.

Comments 3:  Line 229: Please give the explanation for LP abbreviation.

Response 3:  We replaced the abbreviation.

Comments 4: “While the majority of new-borns received appropriate therapy, targeted interventions in antibiotic duration and selection could further optimize neonatal sepsis management”

Response 4: We agree with referee comment, and we changed the discussion as follows: Although targeted interventions in antibiotic duration and selection could further optimize neonatal sepsis management, a significant number of empirical therapies deviated from recommended guidelines.

Round 2

Reviewer 1 Report

Comments and Suggestions for Authors

It has been improved a lot. But still need to modify the discussion part. 

Comments on the Quality of English Language

Need professional English correction. 

Author Response

Comments 1: It has been improved a lot. But still need to modify the discussion part. 

Response 1: We have included a table in the Methods section to simplify its understanding, and we have also expanded the Discussion as requested. Additionally, the English translation has been reviewed by our trusted translator.

Reviewer 2 Report

Comments and Suggestions for Authors

the revision is satisfactory.

Author Response

Response 1: Thank you. We have included a table in the Methods section to simplify its understanding, and we have also expanded the Discussion as requested by the other referees.